

# Data leakage detection in machine learning code: transfer learning, active learning, or low-shot prompting?

Nouf Alturayeif[1,2] and Jameleddine Hassine[1,3]

[1] Information and Computer Science Department, King Fahd University of Petroleum and Minerals, Dhahran, Saudi Arabia
[2] Computing Department, Imam Abdulrahman Bin Faisal University, Dammam, Saudi Arabia
[3] Interdisciplinary Research Center for Intelligent Secure Systems, King Fahd University of Petroleum and Minerals, Dhahran, Saudi Arabia

## ABSTRACT

With the increasing reliance on machine learning (ML) across diverse disciplines, ML code has been subject to a number of issues that impact its quality, such as lack of documentation, algorithmic biases, overfitting, lack of reproducibility, inadequate data preprocessing, and potential for data leakage, all of which can significantly affect the performance and reliability of ML models. Data leakage can affect the quality of ML models where sensitive information from the test set inadvertently influences the training process, leading to inflated performance metrics that do not generalize well to new, unseen data. Data leakage can occur at either the dataset-level (*i.e.*, during dataset construction) or at the code-level. Existing studies introduced methods to detect code-level data leakage using manual and code analysis approaches. However, automated tools with advanced ML techniques are increasingly recognized as essential for efficiently identifying quality issues in large and complex codebases, enhancing the overall effectiveness of code review processes. In this article, we aim to explore ML-based approaches for limited annotated datasets to detect code-level data leakage in ML code. We proposed three approaches, namely, transfer learning, active learning, and low-shot prompting. Additionally, we introduced an automated approached to handle the imbalance issues of code data. Our results show that active learning outperformed the other approaches with an F-2 score of 0.72 and reduced the number of needed annotated samples from 1,523 to 698. We conclude that existing ML-based approaches can effectively mitigate the challenges associated with limited data availability.

# INTRODUCTION

With the advent of digital transformation, the integration of machine learning (ML) code has become widely used across a wide range of disciplines. From biology to finance, engineering to art, practitioners from every discipline are using ML to get new insights, automate complex processes, and innovate. As experts and novices both engage in this complex area, the absence of standardized practices and the complexity of machine learning, often lead to the creation of low-quality code, such as the lack of documentation

Corresponding author
Nouf Alturayeif,
g201901790@kfupm.edu.sa

(*Yang et al., 2021*) and irreproducible code (*Wang et al., 2020a*). Furthermore, researchers from different disciplines write machine learning code that violates best practices, that is continually copied and cloned (*Koenzen, Ernst & Storey, 2020*; *Chattopadhyay et al., 2020*; *Yang et al., 2021*; *Pimentel et al., 2019*). In addition, adversarial attacks pose a unique challenge by exploiting vulnerabilities in ML models through precise and deliberate manipulations of input data, potentially causing the models to make incorrect predictions or behave unpredictably (*Goodfellow, Shlens & Szegedy, 2015*; *Ren et al., 2020*). Several studies proposed methods to improve the quality of ML code, such as generating documentation for data wrangling code (*Yang et al., 2021*), enhancing the reproducibility of Jupyter notebooks (*Wang et al., 2020a*), assessing the best practices of collaborative notebooks (*Quaranta, Calefato & Lanubile, 2022*), and adversarial defenses (*Xie et al., 2019*). Low-quality ML code can lead to a cascade of issues, including increased maintenance costs, decreased system reliability, hindered innovation, and poor quality of the model's predictions.

The accuracy of an ML model in *production* is considered a common quality metric (*Nahar et al., 2022*), providing a real-world measure of the model's performance and its ability to make accurate predictions on new and unseen data under operational conditions. At times, an ML model exhibits satisfactory performance on test data but experiences a significant decline in effectiveness when deployed in a production environment. One potential factor contributing to this scenario is the incorporation of test data (leaked) during the training process, causing the model to overfit to the test set. This results in overly optimistic accuracy estimates (*Yang et al., 2022*) that may not generalize well to new, unseen data. This phenomenon is commonly referred to as *data leakage* (*Burkov, 2020*; *Kaufman et al., 2012*).

While data leakage represents an unintentional issue originating from poor data handling practices, it is useful to contrast this with adversarial attacks, which exploit ML models in a different way. Unlike data leakage, which inflates model performance during training and validation due to unintentional biases, adversarial attacks directly target the model during deployment by intentional manipulation of input data, potentially compromising its robustness and security (*Goodfellow, Shlens & Szegedy, 2015*). Although adversarial attacks and data leakage affect ML systems differently, both highlight the need for robust practices and detection/defense mechanisms to ensure the quality and reliability of ML models across their lifecycle.

Few studies proposed methods to detect data leakage in ML code (*Kaufman et al., 2012*; *Yang et al., 2022*; *Chorev et al., 2022*). However, those studies employed manual detection and code analysis methods. Manual approaches are prone to human error and are time-consuming. In addition, despite the effectiveness of code analysis approaches, they are laborious, as each type of data leakage necessitates a tailored code analysis approach, requires specialized expertise, and may prove challenging (or impossible) to implement for more complex data leaks. Although ML techniques have considerably improved in recent years (*e.g.*, Bidirectional Encoder Representations from Transformers (BERT) (*Devlin et al., 2019*) and Generative Pre-training Transformer (GPT) (*OpenAI, 2023*)), to the best

of our knowledge, the application of ML has not yet been attempted to detect data leakage in ML pipelines.

Nevertheless, training ML models often demands large amounts of annotated data, which is a resource-intensive process that can be a bottleneck in many applications. The conventional paradigm relies on large, annotated datasets, which may not always be available, posing a challenge for certain domains or applications where acquiring extensive labeled data is resource-intensive or impractical. However, recent approaches have emerged to address this challenge. Transfer learning is one approach that leverages pre-trained models on large datasets for related tasks, allowing the model to transfer its knowledge to the target task with limited labeled data (*Zhuang et al., 2020*). In addition, Active learning optimizes the labeling process by selectively choosing the most informative instances for annotation, leading to maximizing the utility of each labeled sample (*Settles, 2009*). Lastly, low-shot prompting is a technique that involves providing explicit examples to guide the model's learning process with minimal labeled data (*Wang et al., 2020b*). These are different strategies that aim to overcome the labeled data availability issue, enhancing the efficiency and effectiveness of training ML.

In this work, we aim to explore the three different approaches to build ML models for data leakage detection in ML code using limited annotated datasets. These models will be trained on code snippets labeled as: (1) have a data leakage, or (2) does not have a data leakage.

In this article, we make the following contributions:

1. Build and annotate a dataset for leakage detection in Python ML code that consists of 1,904 samples.
2. Introduce an automated approach for code augmentation to address the imbalance issue.
3. Investigate the effectiveness of three different ML approaches for limited annotated datasets to detect data leakage in ML code. The three approaches are transfer learning, active learning, and low-shot prompting.
4. Publicly release the dataset and source code.

## RESEARCH BACKGROUND

In this section, we provide an overview of the background related to data leakage, followed by a brief introduction to machine learning approaches tailored for limited annotated datasets.

### Data leakage

Data leakage occurs when testing data are leaked (directly or indirectly, deliberately or unintentionally) to the training process, leading to unrealistically optimistic performance. It can occur because of three common sources: overlap leakage, multi-test leakage, preprocessing leakage (*Burkov, 2020*; *Subotić, Milikić & Stojić, 2022*). In what follows, we briefly describe each type with an illustrative example:

- **Overlap leakage:** Occurs when test data is directly used for training or hyperparameter tuning. In some cases, test data is mistakenly used in creating the training data, such as in augmentation methods. For instance, in the example shown in Fig. 1A, SMOTE oversampling (smote.fit_resample) is applied to the entire dataset (X_resampled, y_resampled) *before* splitting into training and testing sets (line 6). This leads to overlap leakage, as oversampled data in the training set may include information derived from the testing set, compromising the evaluation process.

- **Multi-test leakage:** Occurs when test data is used repeatedly for evaluating the model and making decisions such as: algorithm selection, model selection, and hyperparameter tuning. Instead, validation data should be used. Fig. 1B demonstrates this type of leakage, where cross-validation (cv=RepeatedKFold) is performed on the entire dataset (line 2) during hyperparameter tuning with GridSearchCV (line 11). This setup includes the test set in determining the optimal hyperparameters, indirectly introducing information from the test data into the model tuning process and resulting in multi-test leakage.

- **Pre-processing leakage:** Occurs when test data is merged with the training data for pre-processing. For example, feature selection, normalization, and projection with principal component analysis (PCA). An example is shown in Fig. 1C, where the MinMaxScaler is applied to the entire dataset (line 6) before the data is split into training and testing sets (line 9). This results in preprocessing leakage, as the scaling operation allows information from the test set to influence the training process, leading to overly optimistic performance estimates.

## Machine learning approaches for limited data

As discussed in the introduction, collecting large labeled datasets for machine learning poses significant challenges, impacting various domains and applications. One major obstacle lies in the resource-intensive nature of the process, requiring substantial time, human effort, and financial investment. The need for domain-specific expertise to accurately label data adds another layer of complexity, particularly in specialized fields where domain knowledge is crucial. Additionally, data availability may be limited, restricting the creation of extensive labeled datasets and hindering the development of robust machine learning models. Consequently, addressing these challenges is paramount for advancing the field and fostering the development of more accurate and ethical machine learning applications. In this research, we explore three different approaches, namely, transfer learning, active learning, and low-shot prompting.

### Transfer learning

Transfer learning is an ML paradigm that involves training a model on one task and then transferring its learned knowledge to a different, but related, task. With transfer learning, the model is initialized using the pre-trained weights, and these weights are updated based on the new task-specific small dataset with task-specific objective function, usually with a smaller learning rate (*Liu et al., 2023*).

```
1  # Load the data
2  X, y = load_data()
3
4  # Apply SMOTE to the entire dataset
5  smote = SMOTE(random_state=42)
6  X_resampled, y_resampled = smote.fit_resample(X, y)
7
8  # Split the resampled data into training and testing sets
9  X_train_resampled, X_test_resampled, y_train_resampled, y_test_resampled = train_test_split(
10     X_resampled, y_resampled, test_size=0.2, random_state=42
11 )
12
13 # Train a model on the resampled training data
14 model = RandomForestClassifier()
15 model.fit(X_train_resampled, y_train_resampled)
16
17 # Evaluate the model on the resampled testing data
18 accuracy = model.score(X_test_resampled, y_test_resampled)
```

(a)

```
1  # Create cross-validation splitting strategy
2  cv = RepeatedKFold(n_splits=10, n_repeats=3, random_state=1)
3
4  # Create learning algorithm
5  dtm = DecisionTreeRegressor(random_state=42)
6
7  # Specify hyper-parameters tuning space
8  param_grid = {"criterion": ["mse", "mae"]}
9
10 # Create Grid Search as the tuning strategy
11 search = GridSearchCV(dtm,param_grid, scoring='neg_mean_squared_error', cv=cv)
12
13 # Fit Grid Search on the entire data
14 results = search.fit(X, Y)
```

(b)

```
1 # Create a sacling object
2 scaler = MinMaxScaler()
3
4 # Fit the sacling object and transform the entire data
5 selected_columns = ['median_income', 'total_rooms','population','median_age']
6 SC = scaler.fit_transform(data[selected_columns])
7
8 # Split the data into training and testing
9 x_train, x_test, y_train, y_test = train_test_split(SC, data['median_house_value'], test_size=0.20)
```

(c)

**Figure 1  Data leakage examples.** (A) Overlap leakage, (B) multi-test leakage, (C) pre-processing leakage.

Transfer learning can be categorized into three main types (*Pan & Yang, 2009*; *Zhuang et al., 2020*):

- Inductive transfer learning: It involves transferring knowledge from one task to another, even when the domains are different. This approach requires labeled data in the target domain to train a predictive model for that specific task. For example, a model trained to classify animals could be fine-tuned with labeled car images to perform vehicle classification.
- Transductive transfer learning: It applies when the tasks are the same, but the domains differ. In this scenario, a large amount of labeled data is available in the source domain, but none in the target domain. For instance, a spam email classifier trained on English emails can be adapted to classify spam in French emails.
- Unsupervised transfer learning: It focuses on unsupervised tasks like clustering or dimensionality reduction, where the target task is related to but distinct from the source task. Unlike inductive learning, neither the source nor target domains have labeled data. An example could be using a model pre-trained on a large dataset of generic product reviews (source domain) to cluster customer feedback about a new software product (target domain) into groups even though no labeled data is available in either domain.

Transfer learning has proven to be highly useful and powerful, particularly in deep learning applications, and it has been widely adopted across various domains and tasks (*Zhuang et al., 2020*). In addition, it has been successfully applied across different data types, such as using BERT for text (*Devlin et al., 2019*), ResNet for images (*He et al., 2016*), and Wav2Vec for speech (*Schneider et al., 2019*). Its ability to leverage pre-trained models reduces the need for large labeled datasets in the target domain, making it especially powerful for resource-constrained tasks. For a comprehensive overview, readers are referred to *Zhuang et al. (2020)* and *Pan & Yang (2009)*.

### Active learning

Active learning is a sub-field of ML that produces models of high performance while reducing manual labeling efforts (*Settles, 2009*). A key objective of active learning is to select the most informative data for labeling, with the notion that if the model selects its own data, it will perform better with less training (*Settles, 2009*). It involves an iterative process where the model is trained on a small initial dataset, and then the most informative samples are selected to be labeled. It relies on a *query* function that calculates scores for each data point that needs to be labeled (*Settles, 2009*).

Several strategies for selecting the most informative samples have been proposed (*Settles, 2009*):

- Uncertainty sampling: The model selects instances for which it has the least confidence in predictions. For example, in a binary classification task, this could mean selecting data points where the predicted probability is closest to 0.5.

- Query by committee (QBC): A committee of models with diverse hypotheses selects samples based on the level of disagreement among their predictions (*Seung, Opper & Sompolinsky, 1992*).
- Diversity-based sampling: Ensures the selected samples represent a wide range of the input space, reducing the risk of redundant or overly similar samples being labeled.

Active learning has been applied successfully across various domains. For instance, in medical imaging, models can select ambiguous X-rays for expert annotation, minimizing the workload for radiologists while maintaining diagnostic accuracy. In natural language processing (NLP), active learning helps text classification by identifying and labeling the most uncertain sentences. Similarly, in autonomous driving, it selects edge cases like unusual objects or weather conditions for manual annotation, ensuring robust performance in diverse scenarios.

For readers seeking to explore active learning in depth, comprehensive surveys are available. Settles' foundational survey (*Settles, 2009*) provides an excellent starting point, covering core strategies, theoretical foundations, and practical applications. More recent reviews, such as those by *Ren et al. (2021)*, focus on integrating active learning with deep learning, tackling challenges like scalability and handling high-dimensional data. Domain-specific reviews, such as those in NLP (*Olsson, 2009*), computer vision (*Cohn, Ghahramani & Jordan, 1996*), and medical imaging (*Smailagic et al., 2018*), further highlight its versatility and impact across fields.

### Low-shot prompting

Low-shot prompting refers to another ML paradigm where a model is trained given only a few labeled examples of each class (*Wang et al., 2020b*). Instead of fine-tuning on a task-specific dataset, low-shot prompting relies on providing prompts (or examples) during the inference phase. This approach is particularly useful when only a small number of examples are available for each class, and the model needs to adapt quickly to new tasks. There are typically three low-shot prompting scenarios that describe the amount of examples provided to the model:

- *Zero-shot*: The model is required to perform the task without being provided any task-specific examples. Instead, the model relies on general knowledge learned from pre-training or other tasks to make predictions.
- *One-shot*: The model is provided with only one example per class. The goal is to enable the model to generalize from this single example and make predictions for new instances.
- *Few-shot*: The model is provided with a small number of examples per class. The number of examples is higher than one but still limited, allowing the model to learn from a small amount of task-specific data.

## LITERATURE REVIEW

In this section, we present the state-of-the-art studies on data leakage detection and avoidance in machine learning code, respectively.

Few studies proposed manual approaches for data leakage detection and avoidance (*Kaufman et al., 2012*; *Kohavi & Parekh, 2003*), such as exploratory data analysis (EDA) (*Tukey, 1977*). EDA is an approach to data analysis in which data is explored using a variety of statistical and visual techniques, such as histograms and correlation analysis, in order to gain insight into the structure and relationships of the data (*Tukey, 1977*). One can use EDA to detect *surprising* cases such as unexpected behavior of a feature in a fitted model or surprising model performance. However, there is no doubt that the implementation and execution of manual approaches present greater challenges compared to their automated counterparts. Automated approaches, such as ML-based systems, can exhibit a high level of efficacy in detecting data leakage, making them a better alternative for detecting and avoiding data leakage. Furthermore, automated approaches are also more cost-effective and easier to maintain than manual approaches.

Code analysis (*Yang et al., 2022*; *Drobnjakovic, Subotic & Urban, 2024*; *Cousot & Cousot, 1977*; *Subotić, Milikić & Stojić, 2022*; *Chorev et al., 2022*) is one of the most commonly used approaches for data leakage detection. *Yang et al. (2022)* developed a static data-flow analysis to detect three types of data leakage in ML code: overlap, multi-test, and preprocessing leakage. Their approach tracks the flow of data and detects common patterns that can result in data leakage. For example, multi-test leakage will be reported when only validation data is detected and no testing data is present. On the other hand, overlap leakage will be detected when the model's testing/validation data overlaps with the training data. Lastly, preprocessing leakage will be reported when the training data includes reduced information from the testing/validation data. They found that their analysis accurately detects data leakage with an accuracy of 92.9%. In addition, they found that there is a significant amount of leakage (30%) among over 100,000 public notebooks. *Drobnjakovic, Subotic & Urban (2024)* proposed static code analysis based on abstract interpretation (*Cousot & Cousot, 1977*) that derives an abstract data leakage semantics systematically and rigorously. As an example, when a variable is passed to a function that trains or tests a model, the variable is asserted to be disjoint and untainted. They evaluated their approach in terms of performance and accuracy on 2,088 real-world notebooks. The results show that the approach detects 30 real data leakages with a precision of 94%, while scaling to the performance constraints of interactive notebook environments. *Subotić, Milikić & Stojić (2022)* introduced a static code analysis framework that is specific to notebooks and based on what-if analysis on notebook actions, such as cell executions, creation, and deletion. For example, the framework will warn the user that if a specific cell *A* is executed, data leakage can occur once cells *B* and *C* are executed. On the other hand, the framework can recommend to the user to execute cell *C* then cell *B* in order to avoid data leakage. *Chorev et al. (2022)* employed dynamic code analysis to develop Deepchecks, a Python library to validate different aspects of machine learning models, including overlap data leakage. Nevertheless, their work does not reveal any technical information

about their approach that supports scientific research, instead, it shows information about the tool's capabilities and usage.

While code analysis methods have proven effective for detecting predefined patterns of data leakage, they often struggle with scalability and adapting to complex or novel scenarios in evolving codebases (*Pujar et al., 2024*). In contrast, ML models, such as CodeBERT and GPT, can scale efficiently to large codebases and adapt to diverse coding practices that rule-based methods may miss (*Pujar et al., 2024*; *Zhuang et al., 2020*). Additionally, techniques like transfer learning and active learning further enhance ML-based methods by reducing training overhead and minimizing data annotation requirements (*Pan & Yang, 2009*; *Settles, 2009*), making them a promising alternative for data leakage detection in dynamic software environments.

Few studies proposed approaches to avoid data leakage. *Kaufman et al. (2012)* analyzed data-level leakage in terms of the relationship between inputs ($x$) and target ($y$) samples. They introduced a prevention approach, called *learn-predict separation*, based on analyzing two sources of leakage: features and training samples. The approach consists of two stages: (1) tagging every sample by "is $x$ legitimate for inferring $y$", and (2) only including the features that are purely legitimate for predicting $y$ and only include the inputs that are purely legitimate with all targets as training samples. *Lyu et al. (2021)* studied the data leakage challenges in the context of AIOps (Artificial Intelligence for IT Operations). They found that a time-based splitting of the dataset can significantly reduce the possibility of having data leakage.

Several studies have explored approaches for assisting data scientists in developing ML code that is of higher quality, such as developing frameworks for ML pipelines. Few of these studies can be utilized as a means to detect/avoid data leakage in ML models. For example, *Biswas, Wardat & Rajan (2022)* performed a comprehensive study to understand the nature of data science pipelines in order to facilitate research and practice on the pipelines. Using data science pipelines containing stages of sourcing, cleaning, splitting, normalization, and training can eliminate the need to perform a normalization step before splitting (a type of data leakage). *Namaki et al. (2020)* introduced the *ML provenance tracking* problem, which identifies the columns in a dataset that have been used to train a given ML model. When combined with dependency graphs, data provenance techniques can be used to detect data leakage. The effectiveness of these approaches, however, has not yet been assessed on data leakage detection/avoidance.

Based on the conducted literature review, we identified a gap in applying ML for data leakage detection. In this work, we will fill this gap by proposing ML-based approaches that are generic, scalable, and can be easily extended to any type of data leakage. Additionally, automated ML approaches do not require the same level of manual labor or expertise, making them more accessible and easier to maintain. Exploring ML-based approaches for automation is becoming increasingly popular, as it has the potential to provide more accurate results efficiently and effectively, especially with the breakthroughs in ML-based approaches. Unfortunately, the surveyed approaches did not offer replication packages preventing us from thoroughly evaluating them.

# PROPOSED DATA LEAKAGE DETECTION MODELS

In this research, we examine transfer learning (*Zhuang et al., 2020*), active learning (*Settles, 2009*), and low-shot prompting (*Wang et al., 2020b*) paradigms for data leakage detection as solutions for limited annotated datasets. We utilize CodeBERT (*Feng et al., 2020*) for transfer learning and active learning and GPT (*OpenAI, 2023*) for prompting. We evaluate the effectiveness of the three approaches in terms of recall, precision, and F2-score.

To address the objectives of our study, we aim to answer the following research questions:

- **RQ1:** How effectively can transfer learning identify data leakage in ML code?
- **RQ2:** How does active learning affect transfer learning performance when using a smaller number of training examples?
- **RQ3:** Can low-shot prompting outperform transfer learning in zero-shot, one-shot, and few-shot learning scenarios?

## Data leakage dataset

In this research, we created a dataset that consists of 1,904 labeled samples for data leakage detection. The positive samples (*i.e.*, contain data leakage) constitute 6% of the dataset with 115 samples, whereas the negative samples (*i.e.*, do not contain data leakage) constitute 94% of the dataset with 1,789 samples.

The core of our dataset is Code4ML (*Drozdova et al., 2023*) which contains 7,944 Python code snippets that are publicly available in Kaggle (http://kaggle.com); a platform that is widely recognized as a prominent host for data science competitions. Python was selected as it is one of the most widely used programming languages in machine learning and data science, making it highly relevant for data leakage use cases. Code4ML is manually annotated with the main phases of ML pipelines based on a Machine Learning Taxonomy Tree (*Drozdova et al., 2023*). The taxonomy has two levels, where the high level consists of nine main phases: data export, data extraction, data transformation, debug, environment, exploratory data analysis, hyper-parameters tuning, model evaluation, model interpretation, model training, and visualization. Each high-level phase consists of many low-level phases, resulting in ≈ 80 low-level categories.

The dataset was created in four stages as visualized in Fig. 2.

### *Identify data leakage types*

As a first step, we surveyed the literature to identify the common types of data leakage. As a result, we found that data leakage can occur because of three common sources: overlap leakage, multi-test leakage, and preprocessing leakage (*Burkov, 2020*; *Subotić, Milikić & Stojić, 2022*).

### *Mapping*

In this step, we mapped each data leakage type with the main high-level phases from the taxonomy in which each data leakage may occur. The mapping process is formalized by

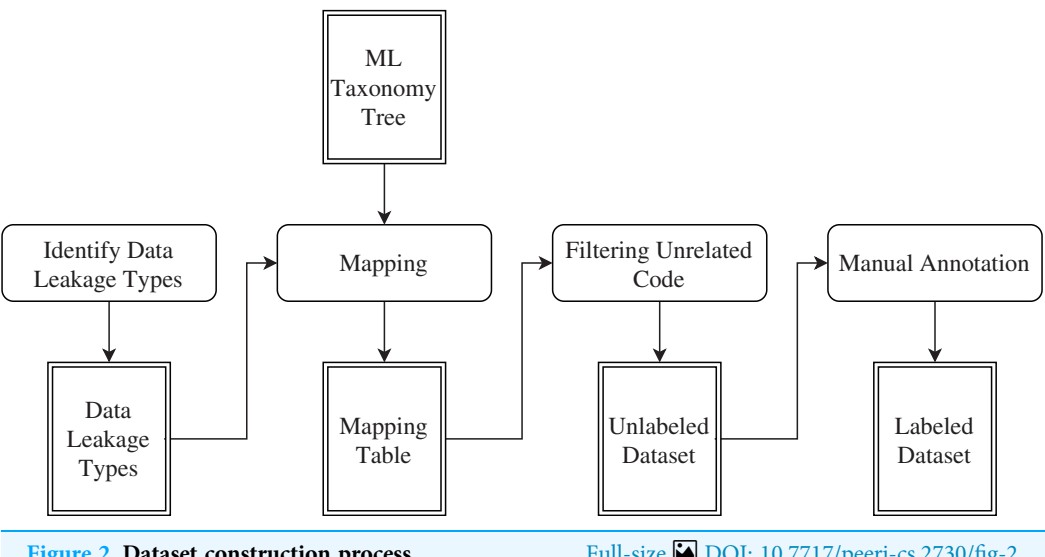

**Figure 2 Dataset construction process.**

**Table 1 Mapping between data leakage types and ML pipeline phases.**

| ML pipeline phase | Leakage types | | |
|---|---|---|---|
| | Overlap | Multi-test | Pre-processing |
| Data export | | | |
| Data extraction | | | |
| Data transformation | X | | X |
| Debug | | | |
| Environment | | | |
| Exploratory data analysis | | | |
| Hyper-parameters tuning | X | X | |
| Model evaluation | | | |
| Model interpretation | | | X |
| Model training | X | X | |
| Visualization | | | |

associating each data leakage type $L$ with the corresponding ML pipeline phases $P$ where the leakage could occur. This can be expressed mathematically as:

$$M = \{(L_i, P_j)|P_j \in \text{Phases}, L_i \in \text{Leakage Types}, f(P_j, L_i) = \text{True}\}$$

where $f(P_j, L_i)$ is a mapping function that evaluates whether a phase $P_j$ is associated with a leakage type $L_i$ based on domain knowledge or observed practices. The mapping results are presented in Table 1.

### Filtering unrelated code

In this step, we used the mapping table to filter Code4ML to include only the code snippets that are associated with a phase that has the potential to introduce data leakage. This step was performed to mitigate the imbalance issue caused by the high volume of unrelated

code. We define "unrelated code" as a code snippet that poses no risk of causing data leakage, such as visualization code. The filtering process can be represented as:

$$C_{\text{filtered}} = \{c \in C | \exists L_i \text{mapped to the phase of } c\}$$

where $C_{\text{filtered}}$ represents the subset of code snippets relevant to data leakage, $C$ is the original set of code snippets, and $L_i$ refers to the identified leakage types. Only snippets associated with phases mapped to at least one leakage type were retained, significantly reducing the dataset size and mitigating the imbalance caused by unrelated code.

Although the unrelated code snippets have been filtered out, the dataset is still imbalanced, with only 4% of the dataset contain code snippets with data leakage. Handling the imbalance issue is discussed in "Data Imbalance" section.

### Manual annotation

In this step, we manually annotated the resulting dataset for data leakage detection. Each code snippet was labeled with either a positive class (contains data leakage) or a negative class (does not contain data leakage). The labeling process was conducted by one author and subsequently validated by an expert in machine learning with a Ph.D. in Computer Science.

## Data imbalance

Data imbalance is characterized by a disproportionate distribution of samples across classes, resulting in biased models that favor the majority class while neglecting the minority class. Training a model on an imbalanced dataset will likely result in suboptimal performance, especially for instances belonging to the minority class. For example, in our study, the minority class comprises only 5% of the dataset (91 samples). When training a model without addressing this imbalance, we observed a high accuracy of 92%, but both precision and recall for the minority class were 0. This indicates that the model predicted all samples as belonging to the majority class, failing to learn patterns from the minority class.

In order to handle data imbalance, oversampling techniques can be used, such as synthetic minority oversampling technique (SMOTE) (*Chawla et al., 2002*) or undersampling techniques, such as random undersampling. These techniques help to mitigate the imbalance of classes, resulting in more accurate models. However, in this study, undersampling was not considered viable because it would have reduced the majority class to match the minority class, resulting in an extremely small training set with only 182 samples in total. Such a reduction would hinder the model's ability to generalize effectively.

As a result, we considered oversampling techniques to address the imbalance, by allowing the model to better learn patterns from the minority class and improving its overall performance.

SMOTE oversamples the minority class by generating synthetic examples using the K nearest neighbors in the feature space, as represented by the equation:

$$x_{\text{new}} = x + \lambda \cdot (x_{\text{nearest}} - x)$$

where $x_{\text{nearest}}$ is a randomly selected nearest neighbor of $x$ in the feature space, and $\lambda$ is a random value between 0 and 1.

LLMs, such as BERT and GPT, are embedding-based models that learn contextual dense vector representations of words, while SMOTE operates in the input feature space and may not be directly applicable to the embeddings learned by LLMs. However, there are alternative techniques to address class imbalance in NLP tasks for embedding-based models, including data augmentation. One of the most common methods of augmentation of code snippets is *code refactoring* (*Fowler & Beck, 1997*; *Tsantalis et al., 2018*). Refactoring is the process of restructuring code to improve its readability, maintainability, and performance without affecting its external behavior (*Fowler & Beck, 1997*). It can involve renaming variables and functions, moving classes and methods, and restructuring classes.

In this research, we utilized GPT model (*OpenAI, 2023*) to automatically augment the dataset (*i.e.*, generate refactored code snippets from each training sample). We followed OpenAI's prompt engineering guide and best practices (https://platform.openai.com/docs/guides/prompt-engineering) to design the prompt. A prompt consists of three roles: System, User, and Assistant (optionally). The System role is used to give instructions to the model, such as asking the model to adopt a persona and use a specific output format. A User represents the entity that interacts with GPT and asks questions. The Assistant is the large language model (*i.e.*, GPT) and can be optionally used in the prompt, such as in the case of few-shot prompting.

While GPT-based refactoring techniques are effective for generating augmented datasets, challenges such as generating semantically incorrect or syntactically invalid code can arise, particularly in complex code snippets. To address these issues, we instructed GPT to perform specific refactoring techniques considered simple and less error-prone, such as renaming variables, functions, or methods. These transformations are inherently less likely to introduce errors and are well-suited for automated augmentation. We instructed GPT to randomly use a combination of a selected set of techniques from the catalog of refactoring techniques (*Fowler & Beck, 1997*). Among the refactoring techniques, we excluded those specific to Object-Oriented Programming (OOP) (*e.g.*, extract classes) as ML code typically emphasizes functional and modular programming over class-based hierarchies. While OOP is integral to ML frameworks like scikit-learn and TensorFlow, pipeline scripts and experimentation often follow a procedural or functional style, focusing on modular, stateless functions. Applying OOP-specific refactoring would risk introducing incorrect refactoring, as these transformations might not align with the functional and modular characteristics of ML code. In order to verify GPT's understanding of the refactoring techniques, we asked GPT to define each one and verified that the answer was accurate. Following is a list of the considered factoring techniques, along with a description of each one:

- **Rename variable:** changes the name of a variable and all references to it.
- **Extract variable:** declares a new variable and assigns the selected expression to it.

- **Change function declaration:** includes renaming the function, adding a parameter, removing a parameter, and changing the signature.
- **Inline function:** replaces the usage of a method with its body, as well as removing the original declaration of the method.
- **Introduce special case:** adds code to handle special cases, such as Null objects.

Figure 3 shows examples of refactored code snippets generated by GPT. Additionally, Fig. 4 shows an example of the prompt used to generate the refactored code in order to create the augmented dataset.

While our study focused on effective and reliable techniques for addressing class imbalance, such as code refactoring, we acknowledge the importance of systematically comparing various methods, including different oversampling and augmentation approaches. Conducting a detailed comparison to evaluate the impact of these methods on performance remains a valuable direction for future work.

## Transfer learning

As discussed previously, transfer learning is an effective approach to use knowledge from one task to help learning model for another related task (*Zhuang et al., 2020*). Models-like BERT (*Devlin et al., 2019*)-are pre-trained on large language corpora and can be fine-tuned for specific tasks such as sentiment analysis or text summarization. RoBERTa (Robustly optimized BERT approach) (*Liu et al., 2019*) is a pre-trained model that builds upon the foundation of BERT to address some limitations of the original BERT model.

Code has specific structure and semantics that are specific to programming languages, making RoBERTa less suitable for direct application to code-related tasks. As a result, CodeBERT (*Feng et al., 2020*) was introduced in 2022 as a pre-trained model for programming languages (Python, Java, JavaScript, PHP, Ruby, Go) to handle code-related tasks. CodeBERT, like its base RoBERTa architecture, consists of multiple layers of transformer units (*Vaswani et al., 2017*), which are particularly effective for sequence-based tasks. CodeBERT achieved state-of-the-art performance on different code tasks, such as code completion and summarization (*Feng et al., 2020*).

Transfer learning primarily involves two phases:

- **Pre-training:** The primary goal of pre-training is to enable the model to learn generalized representations of the input data. In the case of CodeBERT, this involves exposing the model to a large dataset of source code and training it to understand the context and relationships between different code tokens.
- **Fine-tuning:** The fine-tuning phase aims to adapt the pre-trained model to a specific downstream task. In the case of CodeBERT, this could include tasks like code completion, code summarization, or code search.

In this research, we fine-tuned CodeBERT for data leakage detection task. The input of the fine-tuning process is a code snippet, whereas the output is whether it contains a data leakage or not. To load and fine-tune CodeBERT, we used the Hugging Face library (https://huggingface.co). Specifically, we passed the *microsoft/codebert-base* model to

```
1 def norm(x):
2     return (x - data['mean']) / data['std']
3
4 normed_train_data = norm(OH_X_train)
5 normed_test_data = norm(OH_X_test)
```

(a)

```
1 def normalize(x):
2     return (x - dataset['mean']) / dataset['std']
3
4 normalized_train_data = normalize(one_hot_encoded_X_train)
5 normalized_test_data = normalize(one_hot_encoded_X_test)
```

(b)

```
1 def norm(data, x):
2     mean = data['mean']
3     std = data['std']
4     return (x - mean) / std
5
6 normed_train_data = norm(data, OH_X_train)
7 normed_test_data = norm(data, OH_X_test)
```

(c)

```
1 normed_train_data = ((OH_X_train - data['mean']) / data['std'])
2 normed_test_data = ((OH_X_test - data['mean']) / data['std'])
```

(d)

```
1 def norm(x, data):
2     if 'mean' not in data or 'std' not in data:
3         raise ValueError("Data dictionary does not contain required keys")
4     return (x - data['mean']) / data['std']
5
6 normed_train_data = norm(OH_X_train, data)
7 normed_test_data = norm(OH_X_test, data)
```

(e)

**Figure 3 Refactoring examples using GPT.** (A) Original code, (B) refactoring using: Rename Variable, (C) refactoring using: Change Function Declaration, (D) refactoring using: Inline Function, (E) refactoring using: Introduce Special Case.

*RobertaForSequenceClassification* API, which provides a linear layer on top of model output in order to fine-tune it on the downstream task.

To prepare the dataset for CodeBERT, few pre-processing steps are required, which are:

1. Split each code block into tokens.
2. Add the special token [CLS] to the beginning of the tokenized input. The [CLS] token is used to represent the entire sequence for the top linear layer.
3. Pad/truncate the input based on the maximum length allowed by the model (512 tokens).

```
 1 messages = [
 2   {
 3       "role": "system",
 4       "content": '''
 5       You are a specialist in machine learning.
 6       Data leakage in machine learning code occurs when information from
 7       the test set or the future is used during the training phase. This
 8       leads to overly optimistic model performance estimates, as the model
 9       learns patterns that won't generalize well to new, unseen data.
10       You will be provided with a piece of Python code, and your task is
11       to decide if it contains data leakage or not.
12       Answer with yes or no.'''
13       },
14   {
15       "role": "user",
16       "content":
17         <code block>
18       }
19   ]
```

**Figure 4  An example of the prompt used to generate the refactored code.**

4. Replace the tokens with their IDs according to the CodeBERT token indices.

5. Create the Attention Mask array to indicate the padding tokens.

## Active learning

As discusses previously, the active learning algorithm relies on a query strategy to select the most informative samples to be labeled (*Settles, 2009*). In this research, we will employ the least confidence (LC) query strategy, as it is a powerful strategy for binary classification (*Settles, 2009*). LC is particularly effective for binary classification tasks, as uncertainty is directly captured by the posterior probabilities of the two classes *Settles (2009)*. This simplicity makes LC both computationally efficient and well-suited for identifying uncertain samples in binary classification. Alternative strategies, such as margin of confidence and entropy-based sampling, provide more comprehensive measures of uncertainty by incorporating additional information about the label distribution. However, in binary classification, these strategies often give results similar to LC since uncertainty is inherently represented by the two-class probabilities. LC strategy selects the samples that are least certain as to how they should be labeled, as shown in Eq. (1), where $\hat{y}$ is the label with the highest posterior probability for the model $\theta$.

$$x_{LC}^* = \underset{x}{\text{argmax}}(1 - P_\theta(\hat{y}||x)) \tag{1}$$

*Huang (2021)* provided an open-source implementation of active learning for image classification tasks. We employed their implementation and modified it for parsing, pre-processing, and training code. Additionally, we implemented the augmentation process and integrated it with the active learning implementation. The modified active learning

steps are illustrated in Algorithm 1. The active learning algorithm will initialize the training with $N$ selected samples from the unlabeled pool, update the labeled training dataset with the selected samples, augment the dataset, and train the model. The loop will run until there is no improvement of F2-score. For more generalization and robustness, we have added a patience technique, where the loop will break if the F2-score does not improve for $k$ iterations. We experimented with a number of $N$, and $K$ values and concluded that the values ($N = 50$) and ($K = 5$) was the most adequate.

### Low-shot prompting

As discussed in previously, low-shot prompting consists of three prompting scenarios: zero-shot, one-shot, and few-shot. In this research, we explored all three scenarios by leveraging the power of GPT (*OpenAI, 2023*), following the best practices to design the prompt. We selected GPT due to its state-of-the-art performance in language understanding and in-context learning across diverse domains, including software engineering tasks, without requiring task-specific fine-tuning (*Shin et al., 2023*; *Sridhara, Ranjani & Mazumdar, 2023*). We instructed GPT to act like an ML expert, provided it with a definition of data leakage in ML code, and asked it to decide if a specific code block contains data leakage. Figure 5 shows the prompt used to classify the ML code for data leakage detection.

## EXPERIMENTAL EVALUATION

Our primary objective is to evaluate the performance and behavior of transfer learning, active learning, as well as low-shot prompting. Due to the unavailability of the tools and datasets used in existing approaches (*Yang et al., 2022*; *Drobnjakovic, Subotic & Urban, 2024*; *Cousot & Cousot, 1977*; *Subotić, Milikić & Stojić, 2022*; *Chorev et al., 2022*), we encountered a significant limitation in our comparative analysis. Regrettably, this prevented us from conducting a comprehensive assessment of the performance of our proposed approach in relation to these existing methodologies.

Figure 6 illustrates the experiment process, where the arrow colors indicate the paths of the three approaches (blue for transfer learning, red for active learning, and green for low-shot prompting).

### Experiments setup

The dataset was split with a ratio of 80:20, resulting in 1,523 training and 381 testing samples. Additionally, we employed randomization and stratification during the splitting to ensure that the distribution of labels in the training and testing sets remains representative of the overall dataset. In order to ensure a fair comparison between the models, we used the same training and testing sets. Furthermore, to increase the reliability of our results, each experiment was repeated five times with different seed values. The results reported in this article represent the average of five runs.

Although it is important that the proposed model detects data leakage effectively, it is equally important to ensure that the model itself is leak-free. Specific precautions were implemented in response to each of the data leakage types presented previously:

**Algorithm 1   Active learning algorithm.**

*dataset = Data()*

*net = Net()*

*strategy = Strategy(dataset, net)*

**while** no improvements, with patience $K$ **do**

   *query_idxs = strategy.query(N)*

   *strategy.update(query_idxs)*

   *strategy.augment()*

   *strategy.train()*

**end while**

```
 1 messages = [
 2   {
 3       "role": "system",
 4       "content": '''
 5       You are a specialist in machine learning.
 6       Data leakage in machine learning code occurs when information from
 7       the test set or the future is used during the training phase. This
 8       leads to overly optimistic model performance estimates, as the model
 9       learns patterns that won't generalize well to new, unseen data.
10       You will be provided with a piece of Python code, and your task is
11       to decide if it contains data leakage or not.
12       Answer with yes or no.'''
13       },
14   {
15       "role": "user",
16       "content":
17         <code block>
18       }
19   ]
```

**Figure 5  The prompt used for data leakage detection.**

- Overlap leakage: The data augmentation technique to handle the imbalanced data issues is applied only to the training data.
- Multi-test leakage: Test set was not involved in model's decision-making process.
- Pre-processing leakage: No processing of the data was done before splitting the dataset into training and testing sets.

## Evaluation metrics

In the case of imbalanced datasets, traditional metrics such as accuracy may be misleading because a model can achieve high accuracy by simply predicting the majority class most of the time. Therefore, it is essential to consider alternative evaluation metrics that provide a more adequate understanding of the model's performance. Recall, precision, and F-beta score are appropriate metrics to evaluate binary classification problems for imbalanced

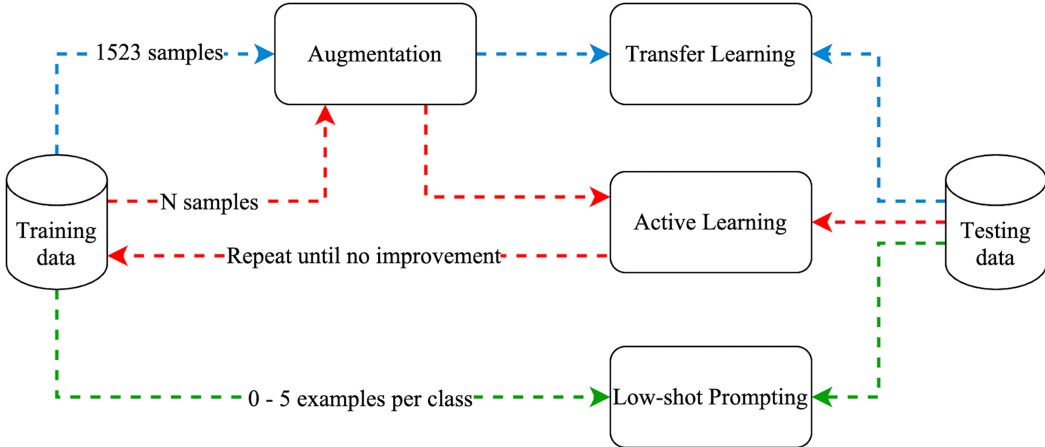

**Figure 6  Experiments methodology.**

datasets due to their sensitivity to minority class identification. In what follows, we describe each evaluation metric:

- **Precision.** Precision is used to measure how many of the detected data leakage samples are correct, using true positives (TP) and false positives (FP) (as in Eq. (2)).

$$Precision = \frac{TP}{TP + FP} \tag{2}$$

- **Recall**: Recall is used to measure how many of the data leakage samples are detected correctly, using TP and false negatives (FN) (as in Eq. (3)).

$$Recall = \frac{TP}{TP + FN} \tag{3}$$

- **F-beta score.** F-beta combines precision and recall into a single value to balance the trade-off between these two metrics (as in Eq. (4)).

$$F\text{-}\beta \text{ score} = (1+\beta^2) \times \frac{Precision \times Recall}{(\beta^2 \times Precision) + Recall} \tag{4}$$

where $\beta$ is a parameter representing the weight assigned to precision compared to recall. For example, when $\beta = 1$, the F-1 score is calculated to provide an equal weight to precision and recall. A $\beta$ value higher than 1 gives more weight to recall, making it useful in situations where recall is more important. Conversely, a $\beta$ value smaller than 1 places more emphasis on precision. In this research, we favor F-2 score (that makes recall twice as important as precision) because recall reflects how many of the actual data leakage samples are detected.

## RESULTS AND DISCUSSION

In this section, we answer the research questions, followed by a discussion on the integration and practicality of the proposed approach.

### RQ1: How effectively can transfer learning identify data leakage in ML code?

To answer this question, we fine-tuned CodeBERT, an LM pre-trained for code-related tasks that is based on the BERT architecture. Additionally, we compared it to a model fine-tuned on the original BERT, that is pre-trained for general NLP tasks. The results are presented in Table 2, where we show recall (R), precision (P), and F-1 and F-2 scores of each run, as well as the average and standard deviation. Our results are align with (*Feng et al., 2020*), as CodeBERT clearly outperform BERT among all metrics.

### RQ2: How does active learning affect the model performance when using a smaller number of training examples?

To answer this question, we evaluated the active learning model (discussed in Algorithm 1) on the testing set. The baseline is the outperforming model resulting from RQ1, which is CodeBERT model fine-tuned on the full dataset (*i.e.*, passive learning). The performance results are shown in Table 3, as well as the number of needed labeled samples and the number of training data (labeled samples + augmented samples). For example, the first run of active learning in Table 3 shows that the model was able to achieve an F-2 score of 0.7 with only 710 labeled samples and a total of 1,219 training samples. It is interesting to note that when comparing the average performance of active learning and passive learning, active learning was able to outperform a model trained on the full dataset (1,523 samples) with only 698 samples. This shows that active learning can lead to better generalization with fewer labeled examples by effectively selecting diverse and informative examples. In consequence, active learning reduces the cost of annotation and thus makes active learning more cost-efficient.

### RQ3: Can GPT outperform the fine-tuned BERT in zero-shot, one-shot, and few-shot learning scenarios?

To answer this question, we performed low-shot prompting on GPT. We experimented with zero-shot, one-shot, three-shot, and five-shot prompting.

The results from low-shot prompting show an increase in performance with additional shots; however, improvement levels off after three-shots (as shown in Table 4). This plateau is consistent with findings from prior studies on few-shot learning, particularly in binary classification tasks. Research suggests that in binary classification, increasing the number of shots often fails to yield further improvements in performance (*Wang et al., 2020b*; *Brown et al., 2020*). This behavior can be attributed to two primary factors. First, in binary classification, the model can often extract sufficient information from a small number of examples due to the relatively simple decision boundary, compared to more complex multi-class tasks (*Wang et al., 2020b*). Additional examples may provide redundant information, limiting further improvements in performance. Second, the ability

**Table 2 Performance of fine-tuning BERT and CodeBERT on data leakage detection task.** Bold values indicate the best performance.

| Model | Run | P | R | F-1 | F-2 |
|-------|-----|-----|-----|-----|-----|
| BERT | 1 | 0.72 | 0.52 | 0.60 | 0.55 |
| | 2 | 0.80 | 0.64 | 0.71 | 0.67 |
| | 3 | 1.00 | 0.36 | 0.53 | 0.41 |
| | 4 | 0.82 | 0.56 | 0.67 | 0.60 |
| | 5 | 0.74 | 0.56 | 0.64 | 0.59 |
| | Avg. | 0.82 | 0.53 | 0.63 | 0.56 |
| | Std. | 0.11 | 0.10 | 0.07 | 0.09 |
| CodeBERT | 1 | 0.89 | 0.68 | 0.77 | 0.71 |
| | 2 | 0.94 | 0.64 | 0.76 | 0.68 |
| | 3 | 0.83 | 0.60 | 0.70 | 0.64 |
| | 4 | 0.76 | 0.64 | 0.70 | 0.66 |
| | 5 | 0.73 | 0.64 | 0.68 | 0.66 |
| | Avg. | **0.83** | **0.64** | **0.72** | **0.67** |
| | Std. | 0.09 | 0.03 | 0.04 | 0.03 |

**Table 3 Performance of active learning approach against models trained on full dataset.** Bold values indicate the best performance.

| Model | Run | P | R | F-1 | F-2 | # Labeled samples | # Training samples |
|-------|-----|-----|-----|-----|-----|-------------------|--------------------|
| Passive learning | 1 | 0.89 | 0.68 | 0.77 | 0.71 | 1,523 | 1,523 |
| | 2 | 0.94 | 0.64 | 0.76 | 0.68 | 1,523 | 1,523 |
| | 3 | 0.83 | 0.60 | 0.70 | 0.64 | 1,523 | 1,523 |
| | 4 | 0.76 | 0.64 | 0.70 | 0.66 | 1,523 | 1,523 |
| | 5 | 0.73 | 0.64 | 0.68 | 0.66 | 1,523 | 1,523 |
| | Avg. | 0.83 | 0.64 | 0.72 | 0.67 | 1,523 | 1,523 |
| | Std. | 0.09 | 0.03 | 0.04 | 0.03 | 0.00 | 0.00 |
| Active learning | 1 | 0.81 | 0.68 | 0.74 | 0.70 | 710 | 1,219 |
| | 2 | 0.90 | 0.72 | 0.80 | 0.75 | 710 | 1,208 |
| | 3 | 0.89 | 0.68 | 0.77 | 0.71 | 770 | 1,461 |
| | 4 | 0.89 | 0.68 | 0.77 | 0.71 | 590 | 866 |
| | 5 | 0.78 | 0.72 | 0.75 | 0.73 | 710 | 1,230 |
| | Avg. | **0.86** | **0.70** | **0.77** | **0.72** | **698** | **1,197** |
| | Std. | 0.06 | 0.02 | 0.02 | 0.02 | 65.73 | 212.69 |

of LLMs to utilize additional examples is constrained by the context window and prompt structure. As the number of examples increases, the signal-to-noise ratio may decline, limiting the model's capacity to extract meaningful patterns (*Brown et al., 2020*). Further improvements might require techniques such as optimizing the selection and diversity of examples or using task-specific fine-tuning to complement prompting approaches.

Zero-shot showed very low performance with an F-2 score of 0.18. One-shot improved the performance with an increase of 0.12 in F-2 score over the zero-shot, and three-shot

**Table 4 Performance of zero, one, and few-shot prompting.** Bold values indicate the best performance.

| Model | Run | P | R | F-1 | F-2 | Model | Run | P | R | F-1 | F-2 |
|-------|-----|------|------|------|------|-------|-----|------|------|------|------|
| 0-shot | 1 | 0.04 | 0.24 | 0.06 | 0.11 | 3-shot | 1 | 0.19 | 0.52 | 0.27 | 0.38 |
|  | 2 | 0.08 | 0.52 | 0.14 | 0.25 |  | 2 | 0.16 | 0.40 | 0.23 | 0.31 |
|  | 3 | 0.06 | 0.40 | 0.10 | 0.18 |  | 3 | 0.21 | 0.60 | 0.31 | 0.44 |
|  | 4 | 0.06 | 0.40 | 0.11 | 0.19 |  | 4 | 0.21 | 0.60 | 0.32 | 0.44 |
|  | 5 | 0.06 | 0.36 | 0.10 | 0.17 |  | 5 | 0.18 | 0.48 | 0.26 | 0.36 |
|  | Avg. | 0.06 | 0.38 | 0.10 | 0.18 |  | Avg. | **0.19** | 0.52 | **0.28** | **0.39** |
|  | Std. | 0.02 | 0.10 | 0.03 | 0.05 |  | Std. | 0.02 | 0.08 | 0.04 | 0.06 |
| 1-shot | 1 | 0.11 | 0.56 | 0.19 | 0.31 | 5-shot | 1 | 0.11 | 0.72 | 0.19 | 0.33 |
|  | 2 | 0.12 | 0.56 | 0.20 | 0.33 |  | 2 | 0.10 | 0.64 | 0.17 | 0.30 |
|  | 3 | 0.09 | 0.44 | 0.15 | 0.25 |  | 3 | 0.09 | 0.48 | 0.14 | 0.25 |
|  | 4 | 0.12 | 0.56 | 0.19 | 0.32 |  | 4 | 0.11 | 0.72 | 0.19 | 0.34 |
|  | 5 | 0.10 | 0.52 | 0.16 | 0.28 |  | 5 | 0.11 | 0.76 | 0.20 | 0.36 |
|  | Avg. | 0.11 | 0.53 | 0.18 | 0.30 |  | Avg. | 0.10 | **0.66** | 0.18 | 0.31 |
|  | Std. | 0.01 | 0.05 | 0.02 | 0.03 |  | Std. | 0.01 | 0.11 | 0.02 | 0.04 |

increased the F-2 score by 0.09 over the one-shot. On the other hand, the five-shot did not improve the overall performance when compared to the three-shot. Figure 7 provides a visual representation of the average performance of the four scenarios. When we increased the shots from three to five, recall increased, but at the expense of precision, F-1 score, and F-2 score. It might indicate that with an increase in the number of shots, the model becomes more biased towards the positive samples (the minority class).

Lastly, we provide a summary of the outperforming models in each of the three proposed approaches: transfer learning, active learning, and low-shot prompting. As shown in Table 5, active learning was the outperforming model with an F-2 score of 0.72. Although prompting requires much less labeled data (*i.e.*, three samples per class), it achieved a low F-2 score value of 0.39. Nevertheless, with active learning, we were able to reduce the number of needed labeled data from 1,523 to 698, while improving performance at the same time. In conclusion, active learning proved to be the most effective method for data leakage detection.

## Practical integration and scalability

Efficient integration of data leakage detection into real-world software engineering workflows is crucial for ensuring continuous code quality and adaptability to dynamic development practices.

Transfer learning utilizes pre-trained models, such as CodeBERT or GPT, which can be fine-tuned on domain-specific datasets with accessible hardware like a single GPU (*e.g.*, NVIDIA RTX 3090 or A100) or cloud services such as AWS EC2. Once fine-tuned, these models are lightweight during inference and can run efficiently on low-cost CPUs. This makes them well-suited for integration into continuous integration/continuous

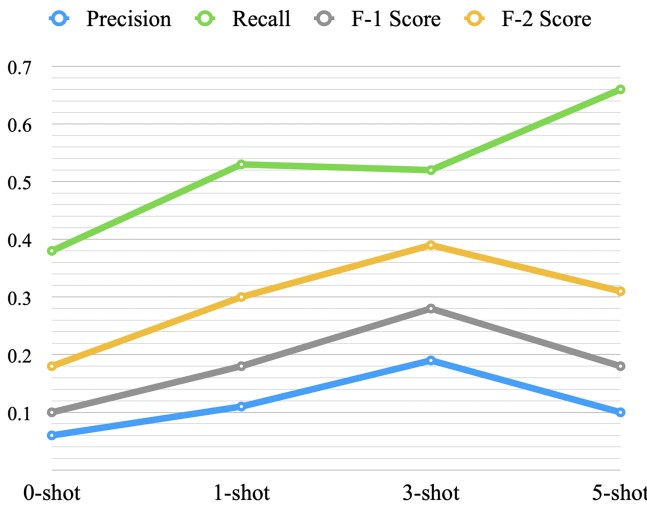

**Figure 7** Average performance of low-shot prompting.

**Table 5 Performance comparison of transfer learning, active learning, and low-shot prompting.**
Bold values indicate the best performance.

| Model | Precision | Recall | F-2 Score |
|---|---|---|---|
| Transfer learning | 0.83 | 0.64 | 0.67 |
| Active learning | **0.86** | **0.70** | **0.72** |
| Low-shot prompting | 0.19 | 0.52 | 0.39 |

deployment (CI/CD) pipelines and code review tools, enabling automated detection of data leakage as part of routine development workflows.

Active learning enhances model performance by focusing on the most informative samples for annotation, reducing the overall labeling effort. This iterative approach allows retraining on incrementally collected data, which can be computationally lightweight and performed on local GPUs or cost-effective cloud infrastructure. By minimizing annotation requirements and computational costs, active learning ensures scalability in rapidly evolving environments. Importantly, even with larger datasets, the number of labeled samples required does not necessarily increase linearly with repository size but is influenced by the complexity and diversity of the codebase as well as the stopping criteria. Future work could explore different sampling methods, such as automated code clustering to group similar code blocks.

To address the iterative nature of code development, where new data leakage issues may arise or previously resolved leaks may reoccur, our approach can be used by incorporating regular model runs into the development cycle. This continuous analysis ensures every code iteration is evaluated for potential leaks. Furthermore, periodic retraining of the model can be incorporated to adapt to significant changes in the codebase, with training efficiency maintained through transfer learning and active learning techniques that minimize computational overhead and data labeling requirements.

## LIMITATIONS

While this research demonstrates promising results, certain limitations must be acknowledged that influence the scope and generalizability of our approach.

First, this study focuses on demonstrating the potential of ML-based approaches for detecting data leakage but does not include a direct comparison with existing state-of-the-art methods, such as static and dynamic code analysis techniques. A comparison was not conducted due to the lack of publicly available benchmarks that comprehensively evaluate these approaches on the same dataset. Furthermore, the surveyed approaches did not provide replication packages, which prevented a thorough evaluation under consistent conditions. Future work should aim to address this limitation by establishing publicly available benchmarks and replication packages to enable comprehensive evaluations of ML-based approaches alongside traditional techniques in terms of accuracy, scalability, and computational efficiency.

Second, the identified types of data leakage do not cover scenarios arising during the data collection and preparation phase, such as when the target variable in a dataset is a function of a feature. This study focuses exclusively on detecting data leakage within the code itself. Additionally, the dataset used in this research is derived from the Code4ML dataset, which consists of isolated code blocks. As a result, our approaches are limited to detecting leakage within a single code block, and future work is required to address leakage that occurs across multiple code blocks or modules.

Third, the study's focus on Python code limits the generalizability of the proposed approach to other programming languages. While Python's prominence in machine learning motivated this choice, further research could explore applying these techniques to additional languages to assess their broader applicability.

Finally, while our transfer learning approach demonstrates the effectiveness of BERT-based models, its generalization to other architectures requires caution. Similarly, the results of low-shot prompting are specific to the GPT model and may not directly translate to other LLMs. Future work could include comparative evaluations across diverse models and architectures to validate these findings further.

## THREATS TO VALIDITY

In this section, we discuss the threats to validity of our study and the steps we took to mitigate them.

One potential threat involves the created dataset, where annotation errors may arise due to the intricate nature of the task or potential inaccuracies in the data entry process. The complexity of data leakage scenarios increases the likelihood of such errors, potentially impacting annotation accuracy. To address this, a rigorous validation process was implemented, including thorough scrutiny by an ML expert.

In the transfer learning approach, we fine-tuned two pre-trained BERT-based models. While our results are specific to these models, they may not generalize to alternative architectures or other BERT-based models. To mitigate this, we selected CodeBERT, which is pre-trained explicitly for code-related tasks, alongside a general BERT-based model, ensuring some diversity in our evaluations.

A potential threat related to the active learning approach is sensitivity to the initial $N$ value, as different values may yield varying results. Lower $N$ values can lead to a cold start problem, where the model struggles to learn due to the limited representation of positive class samples. To mitigate this, we conducted a pilot study to experiment with different $N$ values and selected one that balanced learning performance effectively.

In the low-shot prompting approach, the results are sensitive to the chosen model and prompt design. Different models or prompts might yield better (or worse) performance, potentially affecting generalizability. To address this, we selected the GPT model due to its strong performance in programming-related tasks (*Chang et al., 2024*) and followed OpenAI's best practices for prompt design.

## CONCLUSIONS AND FUTURE WORK

This article highlighted the issue of data leakage in ML code, emphasizing its impact on model performance in production. While existing studies have proposed manual and code analysis approaches to detect data leakage in ML code, the application of ML techniques remains unexplored. In this work, we proposed different ML-based solutions, using transfer learning, active learning, and low-shot prompting, to build effective models for data leakage detection in ML pipelines. These strategies, which mitigate the challenges associated with limited annotated datasets, showed promise in enhancing the efficiency and reliability of ML models. Our experiments showed that active learning outperformed other methods and reduced the amount of labeled data required by half while improving the performance of the model.

Building upon our current exploration of ML approaches for data leakage detection in ML code, future work can be undertaken in many directions. For example, addressing a broader range of data leakage scenarios that may arise in complex ML pipelines. Moreover, it is essential to investigate the impact of code style standardization on model performance. The pre-processing steps could reduce variability in the dataset, potentially improving the model's ability to detect leakage patterns. Additionally, future work could include a comparative evaluation of other LLMs, such as T5 and BART, which support few-shot learning. Additionally, exploring models explicitly fine-tuned on software engineering tasks may provide better performance by capturing domain-specific terms and addressing more complex patterns. Lastly, establishing publicly available benchmarks and replication packages to facilitate the evaluation and comparison of ML-based approaches with state-of-the-art methods. This would provide a deeper understanding of the relative strengths and limitations of these techniques.

## ACKNOWLEDGEMENTS

We used GPT to enhance the writing clarity and readability of the manuscript.

### Funding

This work was supported by the Interdisciplinary Research Center for Intelligent Secure Systems at KFUPM through project No. INSS2406. The funders had no role in study design, data collection and analysis, decision to publish, or preparation of the manuscript.

### Grant Disclosures

The following grant information was disclosed by the authors:
Interdisciplinary Research Center for Intelligent Secure Systems at KFUPM: INSS2406.

### Competing Interests

The authors declare that they have no competing interests.

### Author Contributions

- Nouf Alturayeif conceived and designed the experiments, performed the experiments, analyzed the data, performed the computation work, prepared figures and/or tables, authored or reviewed drafts of the article, and approved the final draft.
- Jameleddine Hassine conceived and designed the experiments, analyzed the data, authored or reviewed drafts of the article, and approved the final draft.

### Data Availability

The data that support the findings of this study are available at Figshare: Alturayeif, Nouf; HASSINE, Jameleddine (2025). Dataset for Data Leakage Detection in Machine Learning Code. figshare. Dataset. https://doi.org/10.6084/m9.figshare.24893799.v1.

The code is available at Figshare: Alturayeif, Nouf; HASSINE, Jameleddine (2025). Code for Data Leakage Detection in Machine Learning Code. figshare. Software. https://doi.org/10.6084/m9.figshare.24893832.v1.

The associated study is available at Drozdova A, Trofimova E, Guseva P, Scherbakova A, Ustyuzhanin A. 2023. Code4ML: a large-scale dataset of annotated Machine Learning code. PeerJ Computer Science 9:e1230 https://doi.org/10.7717/peerj-cs.1230.

The dataset repository is available at Zenodo: Anonymous. (2024). Code4ML 2.0: a Large-scale Dataset of annotated Machine Learning Code (2.0) [Data set]. Zenodo. https://doi.org/10.5281/zenodo.13918465

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
