# Peer review of "Data leakage detection in machine learning code: transfer learning, active learning, or low-shot prompting?"

_PeerJ Computer Science, doi:10.7717/peerj-cs.2730_

## Round 0.1 · original submission · Major Revisions

Please revise the paper according to the reviewer's comments.

Reviewer 1 ·

Basic reporting

no comment

Experimental design

no comment

Validity of the findings

no comment

Additional comments

Following questions are crucial for a more comprehensive understanding of the proposed methods' robustness, practical applicability, and broader impact on machine learning code quality assurance. Addressing them could substantially improve the manuscript's contribution to the field.

1. While the paper discusses the use of oversampling techniques and code refactoring for addressing data imbalance, it lacks a detailed analysis of the impact of these methods on model performance. Specifically, how do different oversampling techniques compare in terms of their influence on accuracy, recall, and F2-score? Could undersampling be viable under different circumstances?
2. The dataset used for training and testing is derived from Code4ML. How well does the proposed model generalize to other datasets or domains beyond Python code, such as Java or C++? Have any experiments been conducted to test cross-domain generalizability?
3. The paper introduces code augmentation using GPT-based refactoring techniques, but does not discuss the challenges associated with this approach. Were there any specific limitations observed during augmentation, such as generating semantically incorrect or syntactically invalid code? If so, how were these limitations managed?
4. Why were certain refactoring techniques excluded from the augmentation process? Would including object-oriented programming-specific techniques (like extract classes) potentially improve the diversity and utility of augmented samples for models trained on object-oriented projects?
5. The paper utilizes the Least Confidence query strategy for active learning. Were any other query strategies considered (e.g., margin sampling or entropy-based)? If so, why was Least Confidence ultimately chosen? If not, would incorporating other strategies yield better performance?
6. The paper evaluates data leakage detection within individual code blocks but does not consider scenarios where leakage may occur across multiple phases of the pipeline. How well would the proposed models perform in detecting leakage that involves complex interactions across multiple code blocks or modules?
7. : The results from low-shot prompting show an increase in performance with additional shots, but at a certain point, improvement levels off. What is the possible reason for this plateau in performance? Does it indicate overfitting to the provided examples, or is there an inherent limitation of GPT in learning more from a limited number of prompts?
8. While the experiments show promising results, how practical is it to apply these techniques in a real-world software engineering workflow? How seamlessly can transfer learning or active learning approaches integrate into existing code review tools or CI/CD (Continuous Integration/Continuous Deployment) pipelines for ML-based systems?
9. The paper mentions the use of ML-based approaches as an alternative to manual and static code analysis methods for detecting data leakage. However, it lacks a comparative discussion on how the proposed methods perform relative to manual or traditional code analysis techniques in terms of effectiveness, scalability, and computational overhead.
10. Machine learning code is often iteratively developed with frequent changes. How does the proposed solution handle such dynamic code changes, where new data leakage issues might be introduced or previously fixed leaks might reoccur? Are there any provisions to retrain the model periodically, and if so, how is training efficiency maintained?
11. The dataset construction process involves manual annotation, which might be influenced by code style variability. How does the proposed model handle variations in coding styles, such as different naming conventions or formatting? Would standardizing code styles prior to training yield better detection performance?
12. The paper utilizes GPT for low-shot prompting. Given that other models (e.g., T5 or BART) also support few-shot learning, was GPT selected based on any comparative analysis? Would a model fine-tuned explicitly on software engineering tasks provide better performance?
13. In the transfer learning approach using CodeBERT, were there any limitations identified during the fine-tuning process, such as challenges in adapting the model to detect subtle patterns of data leakage? How do these limitations compare to those observed with BERT for NLP tasks?
14. Given the large size of modern software repositories, how scalable are the proposed methods, particularly active learning, to large-scale ML codebases? Does the number of required labeled samples increase linearly with repository size, and are there any plans to reduce this requirement further?

·

Basic reporting

Please find below the highlighted comments from reviewing the paper:

- The introduction lacks a distinction between data leakage during training and adversarial attacks in AI models. The authors are encouraged to provide a solid comparison*

- Transfer Learning and Active Learning are briefly discussed in the background section compared to Low-shot Prompting. The authors could provide more detail about types of Transfer Learning and showcase its benefits across different data types, such as speech, images, and video. This should be accompanied by comprehensive reviews and surveys to guide readers to further information. The same recommendation applies to Active Learning.

- In Figure 1, the captions for parts (a), (b), and (c) appear inverted. Please check for consistency and provide more details about data leakage in the caption or in the text where the figure is referenced. Specify the line number, techniques used, functions involved, and relevant arguments in the code that contribute to data leakage.

- Some references are outdated.

- Figures are sometimes referred to as "Fig." and other times as "Figure." Please ensure consistency in labeling.

Experimental design

- The *Data Leakage Dataset* section could be improved by providing equations and mathematical descriptions for some steps, such as mapping, filtering, and SMOTE.

Validity of the findings

- Using only one dataset for experiments is insufficient; at least one additional dataset should be included to ensure the generalizability of the method.

- A comparison with other state-of-the-art data leakage methods is missing.

---

## Round 0.2 · accepted · Accept

Dear Authors,

As the original editor did not respond within the designated timeframe, I have been assigned as the new editor. The two previous reviewers have endorsed your manuscript. I have also conducted a thorough analysis, and I believe that your article has been adequately revised and is ready for publication.

Best wishes,

Reviewer 1 ·

Basic reporting

no comment

Experimental design

no comment

Validity of the findings

no comment

Additional comments

no comment

·

Basic reporting

The paper has been improved considerably

Experimental design

The experiments part is solid

Validity of the findings

The finding is valid